# Quality-Diversity for One-Shot Biological Sequence Design

**Jérémie Donà** [* 1]  **Arthur Flajolet** [* 1]  **Andrei Marginean** [1]  **Antoine Cully** [2]  **Thomas Pierrot** [1]

## Abstract

Designing sequences with desired properties is a common problem in Biology. Relying exclusively on wet-lab experiments to select sequences is costly and time-consuming so in-silico design is often used as a preliminary step. The latter is hard for three reasons. First, the search space is discrete and large. Second, scoring functions quantifying target properties may be inaccurate, especially if fitted on a limited dataset. Third, not all properties can be modeled in silico or measured in vitro, thus requiring in-vivo experiments for evaluation. Strategies have been developed in the literature to address the first two challenges. As for the third one, there is a consensus that concurrently evaluating batches of sequences, supposedly high-performing and diverse, is a good strategy to maximize the chances that at least one design will meet all desidera. Ideally, this is achieved in one shot. We develop a Quality Diversity approach, to guarantee diversity for any batch size. We show that our method outperforms existing ones in terms of diversity, performance, and hyperparameter sensitivity on three datasets from the literature.

## 1. Introduction

The problem of designing biological sequences (e.g. proteins, RNAs, or DNA) with desired properties is commonplace in Biology, in particular in therapeutic applications such as when designing mRNA vaccines (Miao et al., 2021), antibodies with high affinity to disease-associated targets (Liu et al., 2020), or anti-microbial peptides (Fjell et al., 2012). Originally, practitioners relied on a combination of human expert knowledge, heuristics, or random mutations to iteratively select design candidates and evaluate them in expensive and time-consuming wet-lab experiments until one meeting all requirements was found (Cobb

et al., 2013). With the advent of modern Bioinformatics tools, the increased availability of biological data due to improvements in high-throughput assays, and recent advances in AI for protein folding and function prediction (Jumper et al., 2021; Rives et al., 2021), RNA folding (Townshend et al., 2021), or molecular phenotype predictions from DNA sequence (Dalla-Torre et al., 2023), in-silico screening is becoming a trustworthy tool that holds the promise of significant cost savings and shortened development times.

Modern sequence design pipelines are typically broken down in four stages (Jain et al., 2022). In a first stage, in-silico screening is carried out to select a batch of promising design candidates. In a second stage, some of the key desired properties are evaluated experimentally in vitro for all candidates. Finally, sequences that have performed satisfactorily in the in-vitro experiments are further evaluated through in-vivo trials in animals and subsequently in humans. If no single sequence has met all criteria at the end of any stage, the process starts over from the beginning. Given the costs incurred by - and the time spent - preparing and conducting in-vivo (e.g. synthesizing candidate proteins or mRNAs) and in-vitro experiments, it is highly desirable that this be a one-shot process. The exact number of sequences selected as part of in-silico screening is typically much larger than one given the fixed costs of running experiments. We can hope to succeed in a single attempt by leveraging (i) biological modeling and (ii) existing experimental datasets.

One-shot in-silico design of biological sequences is hard for three reasons. First, the search space is discrete, so gradient-based approaches are not readily applicable (Boige et al., 2023), and very large (e.g. around $10^{13}$ points for designing a small protein of 10 amino acids), so exhaustive search is almost never an option. Second, scoring functions may be widely inaccurate for sequences that differ significantly from those in the dataset. Thus, direct optimization of those scoring functions will often yield poor performance in downstream wet-lab experiments (Brookes et al., 2019), especially if the scoring function is a neural network (Szegedy et al., 2013). Third, in-silico screening must output a batch of diverse sequences, in the sense that they cover a variety of modes of the scoring function, to maximize the chances that at least one of them passes all stages. Indeed, since not all desired properties can be estimated in-silico (e.g. toxicity, ability to fold stably, or immunogenicity of antibod-

---

[*]Equal contribution  [1]InstaDeep, Paris, France [2]Imperial College, London, United Kingdom. Correspondence to: Jérémie Donà <j.dona@instadeep.com>.

*Accepted at the 1st Machine Learning for Life and Material Sciences Workshop at ICML 2024.* Copyright 2024 by the author(s).

ies), practitioners avoid undesirable behaviors by sampling several modes of the scoring functions, intuiting that every mode will have distinct hidden characteristics.

Quality Diversity (QD), a sub-field of Optimization, is dedicated to the problem of finding diverse high-performing solutions to an optimization problem. This paradigm has led to breakthroughs over the past decade in many domains ranging from robotics control to engineering design (Gaier et al., 2018; Sarkar and Cooper, 2021; Gravina et al., 2019; Cully and Demiris, 2018). The flagship algorithm of this field, MAP-ELITES (Mouret and Clune, 2015), uses a technique reminiscent of natural evolution. Specifically, MAP-ELITES uses a mapping from solutions to a vector space, the *behavior descriptor space*, to characterize solutions and maintain a data structure, the *repertoire*, filled with high-performing solutions covering this space as much as possible, in a process called *illumination*. Once considered too slow and less sample efficient that gradient based methods, evolutionnary methods benefited from the advent of modern vectorized frameworks such as JAX (Bradbury et al., 2018) and QDAX (Chalumeau et al., 2022), see (Lim et al., 2022).

We bring QD methods, specifically MAP-ELITES, to bear on one-shot biological sequence design problems. Unlike two prior works (Boige et al., 2023; Sambhe et al., 2021), which framework are problem specific and assume that scoring functions are perfectly accurate, we are the first to propose a generic method that can tackle one-shot sequence design. Our contributions are as follows. **1.** Leveraging available datasets, we introduce a generic way to (i) define the behavior descriptor space and (ii) augment the scoring function of MAP-ELITES to constrain the search within trusted regions of the scoring function during optimization. **2.** Within the same evaluation framework used in prior works, we show that our approach outperforms prior ones on three experimental benchmark problems. **3.** We show that our approach is less sensitive to hyperparameters than existing ones.

## 2. Problem Setting

The problem we address is the selection of $N$ sequences $(x_i)_{1 \leq i \leq N}$ in a biological discrete space $\mathcal{X}$ (e.g. $\mathcal{X} = \{1, \cdots, 20\}^K$ for designing a protein with exactly $K$ amino acids without any constraint). The performance of a sequence $x$ is measured by an oracle function $f(x)$ which quantifies how much the designed $x$ meets some of the desired target properties that can be evaluated through in-vivo or in-vitro experiments. The proposed sequences must be high performing respectively to the oracle and diverse in the space of biological sequences. Note that finding a single sequence that maximizes the oracle function is not the final objective as the latter does not capture all desired properties (e.g. toxicity) for the final application. Additionally, the oracle function is not known during the optimization

procedure. Instead, we have access to an offline dataset $(x_i^{\text{ref}}, f(x_i^{\text{ref}}))_{1 \leq i \leq N_{\text{ref}}}$. We assume that biological modeling went in building a parametric family of scoring functions $(\hat{f}_\theta(\cdot))_{\theta \in \Theta}$, typically neural networks, that can be used to approximate $f(\cdot)$ by fitting the parameters $\theta$ on an experimental dataset. Note that, the described problem setting is the same as the one introduced in Trabucco et al. (2021), which was later extended to an active learning setting in Jain et al. (2022) in which sequences are sequentially selected for evaluation in wet-lab experiments in a series of rounds. We use the same metrics defined in Jain et al. (2022); Trabucco et al. (2021) to evaluate the performance, the diversity, as well as the novelty of the selected sequences.

We second Trabucco et al. (2021); Jain et al. (2022), and illustrate the benefits of our method using available experimental datasets of Sample et al. (2019); Sarkisyan et al. (2016); Liu et al. (2020): (i) we fit an expressive neural network on the full dataset which plays the role of the oracle function, (ii) we relabel the dataset with the oracle's predictions since they may not be a perfect reflection of the ground truth values found in the original dataset (we motivate this step in the Appendix), (iii) we hide a fraction of the dataset through imbalanced sampling (high-performing samples are more likely to be hidden than low-performing ones, see appendix E), and (iv) we use fully-connected neural networks as parametric family of scoring functions. Crucially, the oracle and the scoring functions do not share the same neural network architecture and are fitted on different datasets. Any optimization method can be used to select $(x_i)_{1 \leq i \leq N}$ and is evaluated based on: (i) the maximum and (ii) mean oracle performance across the batch, (iii) the average Levenshtein distance between any two sequences in the batch as a measure of diversity, and (iv) the average Levenshtein distance to the offline dataset as a measure of novelty, see appendix E for a mathematical definition of latter metrics.

The objective is to maximize these four metrics. Maximizing the diversity and max (resp. mean) metrics are not necessarily conflicting objectives as diversifying the sequences may help prevent exploiting fitting errors of the scoring functions. Alternatively, one can rely on other techniques to restrict the search space within trusted regions of the scoring functions and select sequences that cover as many of their modes within these regions as possible. One can also use both of these approaches at the same time and this is what we strive to do in this work. Selecting sequences from the offline dataset makes no sense in this framework since those have already been evaluated with the oracle function and methods doing so would get a low novelty metric.

## 3. Method

We use MAP-ELITES, the flagship algorithm of the QD field. Implementations of it differ along: (i) the mutation operators

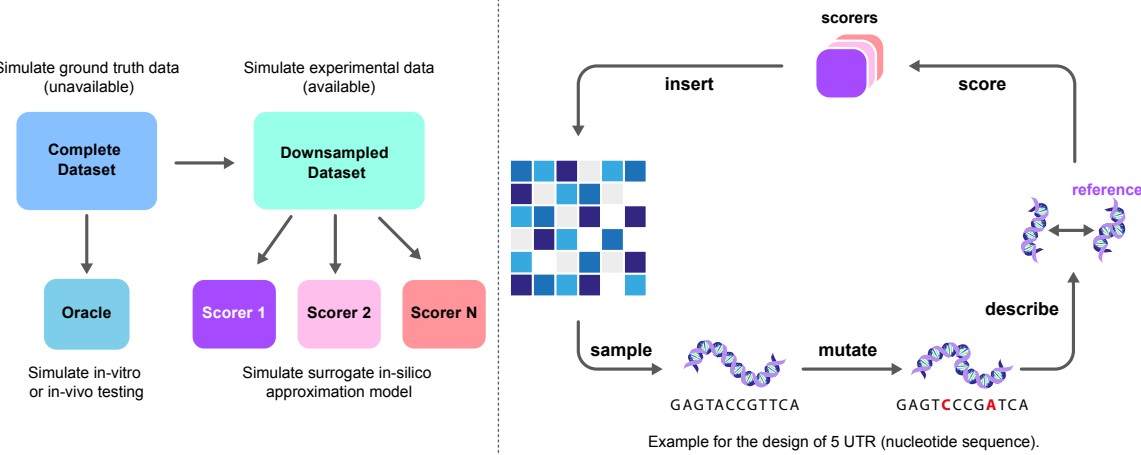

*Figure 1.* Left. Schematic overview of our experimental protocol. An oracle, e.g. an expressive neural network is learned from real data. It enables us to relabel the dataset and emulates wet-lab results. An ensemble of scoring functions are learned from this relabelled dataset. Right. We optimize a MAP-ELITES grid with respect to this ensemble of scoring functions, following eq. (2), and the descriptors of eq. (4)

used to generate new solutions from existing ones, (ii) the definition of the function used for evaluation, referred to as *fitness* function, and (iii) the mapping between solutions to a multi-dimensional space, referred to as *behavior descriptor space*, which we want to cover as fully as possible. In Section 3.1, we give a reminder of the inner workings of MAP-ELITES. In Section 3.2, leveraging the offline dataset and the parametric family of scoring functions, we set the fitness function as a conservative estimate of the oracle function. In Section 3.3, we detail our strategy to ground the behavior descriptor space around the sequences from the offline dataset, allocating less volume to sequences that are far from it. We illustrate our method in Figure 1.

### 3.1. Overview of MAP-Elites

MAP-ELITES takes as an input a function, the *behavior descriptor*, denoted by $b(\cdot)$, mapping solutions to the behavior descriptor space along with a collection of points $(b_j)_{1 \leq j \leq J}$ in this space, denoted *centroids*, that split it into $J$ regions (associating points to the closest centroid), denoted *cells*. Our choices of $b(\cdot)$ and $(b_j)_{1 \leq j \leq J}$ are given in Section 3.3. The algorithm maintains a data structure, *the repertoire*, where one solution can be stored per cell. MAP-ELITES proceeds in iterations where a prescribed number of solutions are: (i) sampled uniformly from the non-empty cells, (ii) mutated and, (iii) inserted into the repertoire according to the following rule: if the cell corresponding to the behavior descriptor of the solution at hand is empty, the solution is assigned to this cell. In the opposite situation, the cell is updated only if the solution has a greater fitness than the current incumbent and is dropped otherwise. To initialize the repertoire, a set of initial solutions are inserted into it using the aforementioned rule. Our mutation operator simply se-

lects a random position to mutate the sequence and replaces it with a random amino acid (resp. nucleotide) for protein (resp. RNA and DNA) applications. Extensive details and MAP-ELITES implementation are given in Appendix C.

### 3.2. Fitness Function

It is well-established that neural networks are susceptible to adversarial attacks either based on gradient or evolutionnary methods (Nguyen et al., 2015; Alzantot et al., 2019). This is problematic in our setting as we are maximizing neural network scoring functions, fitted only on a very small portion of the space of possibles. For instance, the GFP dataset (Sarkisyan et al., 2016) considered in our experiments, contains about 50,000 data points for $\sim 10^{32}$ possibilities (all proteins of length 237 with at most 11 mutations from the wild type). We use two mechanisms to fight this behavior.

First, we use train multiple scoring functions and ensemble their outputs to reduce the prediction error (Lakshminarayanan et al., 2017). This technique is often used for sequence design, such as in Liu et al. (2020); Bryant et al. (2021); Angermueller et al. (2020b) where several models are averaged to optimize the input sequence. However, adversarial examples with largely overestimated scores are still likely to be found since evolutionary algorithms (including MAP-ELITES) require a large number of function evaluations to converge (Lehman et al., 2020). As a result, instead of optimizing the average score, we optimize a high-probability lower bound on the average score given by:

$$s(x) = \frac{1}{L} \sum_{l=1}^{L} \hat{f}_{\theta_l}(x) - \beta . \hat{\sigma}\big((\hat{f}_{\theta_l}(x))_{1 \leq l \leq L}\big), \quad (1)$$

where $(\theta_l)_{l=1,\cdots,L}$ are parameters fitted on the dataset (e.g.

using different initial random values), $\beta$ is a positive hyper-parameter, and $\hat{\sigma}$ is the empirical standard deviation. This enables us to avoid exploiting overestimation errors that occur randomly as a function of the parameters' initial values.

We also augment the scoring functions' loss to penalize overestimation on out-of-distribution (ood) data. We follow the approach developed in Trabucco et al. (2021) which consists in adding a penalty term to the regression loss (e.g. the $\ell_2$ loss). This term penalizes overestimation of the performance on ood samples that are obtained through gradient ascent on the input (as opposed to randomly chosen from the dataset). Mathematically, this term is:

$$L_{robust}(\theta) = \mathbb{E}_{\mu(x)}[f_\theta] - \mathbb{E}_{p_{data}}[f_\theta], \quad (2)$$

where $p_{data}$ is the dataset's distribution and $\mu(x)$ is the distribution of data points obtained by gradient ascent at $x$. We detail this approach in appendix D.

### 3.3. Similarity-Based Behavior Descriptor Function

The behavior descriptor space must be carefully designed to maximize the metrics defined in eq.(6) in our setting. This is because MAP-ELITES strives to cover - a discretization of - this space as fully as possible to maximize the diversity of the solutions it outputs. Thus, naively defining this space may result in allocating a large number of cells to portions of the search space that are far way from the offline dataset distribution, thereby making any strategy discussed in Section 3.2 to penalize overestimation on ood sequences ineffective. Moreover, we need to discretize the behavior descriptor space in a generic fashion that avoids degenerate scenarios where few cells cover the search space. Also, the behavior descriptor space must be low-dimensional, as MAP-ELITES is known to struggle with high-dimensional ones (Colas et al., 2020).

Let $x \in \mathcal{X}$ be a sequence and $\mathcal{X}^{ref}$ be a subset of sequences found in the offline dataset, referred to as *reference sequences* hereafter. In practice, $\mathcal{X}^{ref}$ is chosen to be the full offline dataset except if the latter is large, for computational reasons that will become clear below. Following a long-standing practice of using pairwise sequence similarities in Machine Learning and Computational Biology (Liao and Noble, 2003; Tsuda, 1999), we describe $x$ by: (1) identifying the most similar sequences in $\mathcal{X}^{ref}$, and (2) computing the average similarity to these references. We first compute the pairwise similarities w.r.t. the reference sequences:

$$\phi(x) = \big(\mathrm{H}(x,y)\big)_{y \in \mathcal{X}^{ref}} \in \mathbb{R}^{N_{ref}}, \quad (3)$$

where H is a function measuring the similarity between two sequences. In this work, we use $\mathrm{H}(x,y) = -\mathrm{d}(x,y)$, where $\mathrm{d}(x,y)$ is the Levenshtein distance between $x$ and $y$. We define $\phi_n(x) = \mathtt{softmax}(\phi(x)) \in \mathbb{R}^{N_{ref}})$ and compute

the average similarity of $x$ with the reference sequences as $\mathrm{d}_n(x) = -\frac{\phi_n(x) \cdot \phi(x)}{R} \in \mathbb{R}$, where $R \in \mathbb{R}$ is a normalization scalar which we set to half of the average pairwise distance between any two sequences in the offline dataset. The larger $\mathrm{d}_n(x)$ is in absolute value, the further away $x$ is from $\mathcal{X}^{ref}$. Finally, in a step reminiscent of the use of low-dimensional randomized feature spaces in Machine Learning (Rahimi and Recht, 2007), we project down $\phi_n(x)$ with a random $d \times N_{ref}$ real-valued matrix $W$ in order to build a low-dimensional behavior descriptor vector:

$$b(x) = e^{d_{nn}(x)} W \phi_n(x) \in \mathbb{R}^d. \quad (4)$$

We sample the components of $W$ uniformly over $[0,1]$ as: (1) this eases the control of the volume of behavior descriptor space dedicated to ood sequences, and (2) we can use a generic discretization strategy of $[0,1]^d$ - the Centroidal Voronoi Tessellation (Vassiliades et al., 2017) - to define the behavior descriptor centroids $(b_j)_{1 \leq j \leq J}$. This choice guarantees that $b(x)$ is in $[0, e^{d_{nn}(x)}]^d$ since the components of $\phi_n(x)$ are non-negative and sum to one. Also, $\mathrm{d}_n(x)$ is large and $b(x)$ is far away from the origin if $x$ is far away from $\mathcal{X}^{ref}$ by construction. Conversely, if $x$ is close to $\mathcal{X}^{ref}$, $\mathrm{d}_n(x)$ is close to 1 and the components of $b(x)$ are independent random variables with support in $[0,1]$. Moreover, any two sequences $x,y \in \mathcal{X}$ close to $\mathcal{X}^{ref}$ but very different from each other will have their descriptor behaviors lie in different portions of $[0,1]^d$ with high probability as $\phi_n(x) \cdot \phi_n(y)$ will be close to 0 (i.e. $b(x)$ and $b(y)$ are weighted averages of independent uniform random variables).

## 4. Experiments

We now benchmark our approach against state-of-the-art methods from the literature on experimental datasets spanning three standard applications: optimizing the function of an existing protein (GFP), optimizing a non-coding region of an existing mRNA in order to maximize translation efficiency (5-UTR), as well as designing an antibody with high binding affinity to a specific target (AB), detailed in appendix B. Architectures details for oracles and scoring functions are to be found in appendix E.

**Experimental Framework.** As in Trabucco et al. (2021); Jain et al. (2022), for each of the dataset, optimization methods are required to output a batch of $N = 128$ sequences $(x_i)_{1 \leq i \leq N}$ evaluated using the metrics defined in eq.(6). Given that most methods start from an initial pool of sequences and in order to minimize the comparative performance variance due to the choice of these sequences, all methods use the same 128 starting sequences. For the GFP and 5-UTR datasets, we take the same 128 starting sequences as in Trabucco et al. (2021). For the AB dataset, we take the 128 top scoring sequences. For each dataset, all optimization methods are run 5 times with different random

*Table 1.* Mean and standard deviation of metrics from eq.(6) over 5 runs for various methods on the GFP dataset.

| METHOD | MAX | MEAN | DIVERSITY | NOVELTY |
|---|---|---|---|---|
| CBAS | $0.84 \pm 0.05$ | $0.81 \pm 0.06$ | $2.7 \pm 1.3$ | $0.9 \pm 0.4$ |
| CMA-ES | $0.00 \pm 0.09$ | $-0.19 \pm 0.04$ | $\mathbf{232 \pm 0.4}$ | $\mathbf{200 \pm 1.5}$ |
| COMS | $0.86 \pm 0.00$ | $0.75 \pm 0.00$ | $5.9 \pm 0.0$ | $0.0 \pm 0.1$ |
| GA | $0.86 \pm 0.01$ | $0.80 \pm 0.00$ | $7.9 \pm 0.2$ | $3.7 \pm 0.1$ |
| GFLOWNET | $0.86 \pm 0.02$ | $0.42 \pm 0.16$ | $86.9 \pm 9.9$ | $110.3 \pm 12$ |
| GRAD | $0.86 \pm 0.00$ | $0.75 \pm 0.00$ | $5.9 \pm 0.0$ | $1.1 \pm 0.7$ |
| REINFORCE | $0.83 \pm 0.06$ | $0.71 \pm 0.03$ | $5.9 \pm 0.0$ | $2.1 \pm 0.0$ |
| **OURS** | $0.86 \pm 0.00$ | $\mathbf{0.82 \pm 0.01}$ | $8.5 \pm 0.4$ | $4.3 \pm 0.3$ |
| OURS-BIO | $\mathbf{0.87 \pm 0.00}$ | $0.44 \pm 0.08$ | $8.2 \pm 0.9$ | $8.2 \pm 0.2$ |

*Table 2.* Mean and standard deviation of metrics from eq.(6) over 5 runs for various methods on the 5-UTR dataset.

| METHOD | MAX | MEAN | DIVERSITY | NOVELTY |
|---|---|---|---|---|
| CBAS | $0.69 \pm 0.00$ | $0.55 \pm 0.00$ | $37.1 \pm 0.0$ | $22.5 \pm 0.0$ |
| CMA-ES | $0.71 \pm 0.00$ | $0.66 \pm 0.00$ | $\mathbf{39.6 \pm 0.1}$ | $\mathbf{48.4 \pm 0.1}$ |
| COMS | $0.70 \pm 0.00$ | $0.68 \pm 0.00$ | $36.5 \pm 0.0$ | $1.0 \pm 0.1$ |
| GA | $0.69 \pm 0.00$ | $0.68 \pm 0.00$ | $25.5 \pm 1.9$ | $11.8 \pm 5.4$ |
| GFLOWNET | $0.70 \pm 0.05$ | $0.56 \pm 0.06$ | $27.3 \pm 4.2$ | $21.6 \pm 1.0$ |
| GRAD | $0.70 \pm 0.00$ | $0.65 \pm 0.01$ | $33.9 \pm 0.6$ | $22.4 \pm 0.2$ |
| REINFORCE | $0.68 \pm 0.00$ | $0.53 \pm 0.01$ | $36.9 \pm 0.0$ | $22.6 \pm 0.0$ |
| **OURS** | $0.71 \pm 0.01$ | $\mathbf{0.69 \pm 0.00}$ | $14.7 \pm 2.5$ | $22.0 \pm 0.6$ |
| OURS-BIO | $\mathbf{0.72 \pm 0.00}$ | $0.68 \pm 0.00$ | $35.1 \pm 1.6$ | $21.7 \pm 0.3$ |

seeds, each time regenerating the available offline dataset and refitting the scoring functions.

**Baselines.** We benchmark our approach against state-of-the-art methods as well as a standard baselines. These span the main categories of methods for biological sequence design investigated in the literature: (1) gradient approaches (COMS and GRAD), (2) evolutionary approaches (CMA-ES and GA), (3) generative modeling (GFLOWNET), (4) Bayesian Optimization approaches (CBAS), and (5) Reinforcement Learning approaches (REINFORCE). COMS, developed in Trabucco et al. (2021), consists in running gradient ascent on a scoring function fitted using an augmented regression loss penalizing overestimation on ood samples, see Section 3.2. GRAD consists in running gradient ascent on a scoring function fitted using the $\ell_2$ regression loss. We consider variants of COMS and GRAD that run gradient ascent on ensembles of scoring functions with only little impact on the experimental results, see the Appendix. CMA-ES (resp. RE-INFORCE) refers to the adaptation of the Covariance Matrix Adaptation evolution strategy (Hansen, 2006) (resp. the Reinforce algorithm (Williams, 1992)) to biological sequence design problems from Trabucco et al. (2021). GA is a naive genetic algorithm that randomly mutates the initial pool of solutions a prescribed number of times, evaluates them all with an ensemble of scoring functions, and outputs the $N$ best-performing ones. GFLOWNET (resp. CBAS) refers to the approach developed in Jain et al. (2022) (resp. Brookes et al. (2019)). Finally, we propose an alternate version of our

method that uses biologically-grounded behavior descriptors instead of eq.(4), we derive the mean polarity, the mean net charge, and the mean hydrophobicity of constituent amino acids for protein datasets (namely AB and GFP). We normalize the behavior descriptors using the max and min statistics computed on the training set. We report hyperparameters choices for all methods (including ours) in the Appendix. All experiments are run on a TPU v3-8 with no time limit.

**Performance Analysis.** Results are reported in Tables 1, 2, and 4 for all optimization methods and all datasets. We also depict in fig. 2 the evolution of several QD metrics for the GFP task. Note that extensive ablation studies and hyperparameters senstivity analysis are conducted and reported in appendix G. First, observe that our approach obtains the best performance in terms of the mean metric across datasets while all others underperform by at least 9% on at least one dataset. Second, we argue that our approach outperforms any other single method on at least two datasets out of three. Indeed, CMA-ES and GFLOWNET are outperformed in terms of the mean metrics on the GFP and AB datasets, so much so that the boost in diversity and novelty they yield is irrelevant. In fact, the high diversity and novelty metrics obtained by these two methods on GFP are likely the cause of their poor performances as this suggest that they select sequences far away from the offline dataset distribution. Note that high novelty and diversity scores were also reported for CMA-ES and GFLOWNET in Jain et al. (2022). The novelties of solutions selected by COMS are very low compared to the other methods for the 5-UTR and GFP datasets (less than two mutations on average) and it is thus unclear whether this method yields results that are consistently better than directly selecting sequences from the offline datasets. GRAD is outperformed in terms of the mean metric on the GFP and 5-UTR datasets. GA performs strictly worse than our approach w.r.t. all metrics on the GFP dataset and it is significantly outperformed in terms of the mean metric on the AB dataset, so much so that the boost in diversity it yields is irrelevant. Finally, CBAS and REINFORCE are also outperformed in terms of the mean metric on the 5-UTR and AB datasets. Finally, we present in fig. 3 the repertoire for the anti-body design task showing various excellent high performing areas corroborating the metrics presented in table 4.

## 5. Conclusion

We have demonstrated that QD optimization is a promising tool for the problem of designing diverse and high-performing biological sequences. While we have focused on the one-shot setting, we believe that active learning settings with multiple rounds of evaluations are also of great interest to practitioners, leave this extension for future work.

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

# A. Related Work

Previous works propose various techniques to: (i) learn continuous representations of the sequence space to be able to run gradient ascent (Castro et al., 2022; Gómez-Bombarelli et al., 2018) and (ii) stay within the scoring function's trusted regions of sequence space during optimization (Trabucco et al., 2021; Brookes et al., 2019). While there is a consensus that enforcing diversity in the pool of design candidates is paramount for success (Trabucco et al., 2021; Jain et al., 2022; Angermueller et al., 2020a; Linder et al., 2020; Bryant et al., 2021), diversity measures vary widely across works and most existing approaches induce diversity in an ad hoc fashion (e.g. by using different initial starting points for gradient ascent (Trabucco et al., 2021; Castro et al., 2022)). Approaches that target diversity directly do it through sampling from generative models (Jain et al., 2022; Linder et al., 2020), but diversity is only guaranteed for large batch sizes. A possible remedy for batch sizes in the range of 10 to 100 sequences is to sample many more sequences and use clustering techniques (Bryant et al., 2021) but this would be done in ad hoc fashion that might compromise performance.

**Active learning setting.** Many prior works tackle a setting where batches of sequences are sequentially selected for evaluation in wet-lab experiments in a series of rounds, as opposed to a single round in this work. These approaches are formulated in the fields of Bayesian Optimization (Pyzer-Knapp, 2018; Moss et al., 2020), Reinforcement Learning (Angermueller et al., 2020b), Online Learning (Sinai et al., 2020), Generative Modeling (Jain et al., 2022; Linder et al., 2020), or Population-Based Optimization (Angermueller et al., 2020a). There is a clear exploration-exploitation trade-off to resolve in this case which does not exist in the one-shot setting. Of these works, only Jain et al. (2022) can leverage an offline dataset and evaluate the approach in the one-shot setting.

**Gradient-based methods.** In the hope of exploring the large sequence space efficiently, prior works use gradient-based methods in a closely-related continuous space. This comes with the downside of having to project back onto the sequence space, possibly incurring significant performance loss in the process. Some works run gradient ascent on a simple continuous representation (e.g. one-hot representation in Liu et al. (2020) or log probabilities in Trabucco et al. (2021)). A more involved strategy (Gómez-Bombarelli et al., 2018; Castro et al., 2022; Gómez-Bombarelli et al., 2018) is to train an auto-encoder model alongside the scoring function and feed the latent representation as input to the latter. Gradient ascent is carried out in this latent space. Additional losses are introduced for the encoder's training to regularize this space.

**Restricting the search space.** Multiple orthogonal approaches have been proposed to stay within trusted regions of the scoring function during optimization. Castro et al. (2022); Trabucco et al. (2021) introduce a conservative training procedure for the scoring function. Brookes et al. (2019); Jain et al. (2022); Linder et al. (2020) use generative models trained on the dataset or a set of realistic samples. Kumar and Levine (2020); Chen et al. (2023) learn inverse mappings. Liu et al. (2020) estimate the epistemic uncertainty of the scoring function through ensembling methods.

**Promoting diversity.** Most prior works generate batches of sequences as diverse as possible in an ad hoc fashion. Trabucco et al. (2021); Castro et al. (2022) use various initial sequences for running gradient ascent. Angermueller et al. (2020a) use an ensemble of methods. Sinai et al. (2020) rely on the inherent stochasticity of the optimization strategy. Sampling as many sequences as needed is the default strategy for approaches based on Generative Modeling (Jain et al., 2022), though the model is sometimes explicitly trained to generate diverse sequences (Linder et al., 2020). Boige et al. (2023) use MAP-ELITES to induce diversity but assume that the neural-network scoring function is perfectly accurate. Finally, while we propose a simple QD approach using MAP-ELITES to promote diversity in a custom descriptor space, several works related to the QD field address the promotion of diversity by instead learning the behavior descriptor space Grillotti and Cully (2022); Cully (2019).

# B. Datasets

**Green Fluorescence Proteins (GFP) dataset (Sarkisyan et al., 2016).** The authors of this work run high-throughput assays to study how mutations of the *Aequorea victoria* protein affect its fluroscence ability. The datasets consists of around 50,000 variants obtained through random mutagenesis, with an average (resp. maximum) of 3.7 (resp. 11) mutations compared to the wildtype sequence. All proteins in the dataset are made of 237 amino acids as the authors only consider amino-acid substitutions (i.e. never deletions nor insertions). This dataset has been used extensively in the literature (e.g. in Castro et al. (2022); Jain et al. (2022); Trabucco et al. (2021); Brookes et al. (2019)) to benchmark sequence design methods where the objective is to maximize the amount of light the protein emits. The experimental data was originally collected by

Sarkisyan et al. (2016). We use the dataset extracted from it by the authors of Brookes et al. (2019), just like Trabucco et al. (2021); Jain et al. (2022). This processed dataset is available at `https://github.com/dhbrookes/CbAS/tree/master/data` and normalized so that experimental measurements lie in $[0, 1]$. We split this dataset into a training set and a test set with 90% of the sequences randomly assigned to the former.

**5' Untranslated Transcribed Region (5-UTR) dataset (Sample et al., 2019).** The 5' UTR region refers to the beginning of an mRNA sequence that ends just before the portion encoding a protein. It is well-documented that - though poorly understood how - the content of the 5' UTR region significantly affects the rate at which mRNAs are translated into proteins by ribosomes in eukaryotic cells (Neelagandan et al., 2020). Sample et al. (2019) run massively parallel assays to characterize this dependency for a fluorescent protein. Specifically, they randomly mutate the last 50 nucleotides of the wildtype mRNA's 5' UTR region and measure the resulting average number of ribosomes loaded onto mRNAs in in-vivo experiments for a total of around 280,000 variants. This dataset is often used (e.g. in Trabucco et al. (2021); Angermueller et al. (2020a)) to benchmark design methods where the objective is to maximize the ribosome load (a quantity that positively correlates with translation rate). The experimental data was originally collected and made available by Sarkisyan et al. (2016) at `https://ftp.ncbi.nlm.nih.gov/geo/samples/GSM3130nnn/GSM3130435/suppl/GSM3130435_egfp_unmod_1.csv.gz`. As in Sample et al. (2019); Trabucco et al. (2021); Angermueller et al. (2020a), we sort all sequences in the original dataset by total number of reads and only keep the top 280,000 ones. The experimental measurements are normalized to lie in $[0, 1]$. Out of these sequences, we withhold 20,000 randomly-selected sequences for the test set.

**Antibody Design (AB) dataset (Liu et al., 2020).** The structure of the third complementarity determining region of the antibody heavy chain (CDR-H3) significantly impacts the ability of an antibody to bind to a given antigen. The authors of Liu et al. (2020) run high-throughput phage panning experiments on a libary of around 60,000 CDR-H3 variants of the anti-ranibizumab sequence. The CDR-H3 portions were randomly generated amino-acid portions ranging in length from 10 to 18. The experiments proceeds in multiple rounds and the authors report the measured log ratios of sequence enrichment between rounds of selection, a quantity correlating positively with target binding affinity. This dataset is used in the literature (e.g. in Castro et al. (2022); Liu et al. (2020)) to benchmark design methods where the objective is to optimize the CDR-H3 portion to maximize the binding affinity to the target (measured indirectly via the estimated log ratio of sequence enrichment). The experimental data was originally collected by Liu et al. (2020). In practice, we use the dataset extracted from it by the authors of Castro et al. (2022). Train and test splits for this dataset are available at `https://github.com/KrishnaswamyLab/ReLSO-Guided-Generative-Protein-Design-using-Regularized-Transformers/tree/main/data/gifford_data`.

## C. Additional Details on our MAP-Elites-Based Method

**Mutation operators.** As described in section 3.1, we use the simplest possible mutation strategy for MAP-ELITES. First, we define the maximum number of mutations $M$ that can be applied w.r.t. a base sequence. We chose for this sequence: the wild-type sequence for GFP, the sequence in the experimental dataset with the highest number of reads for 5-UTR, and the sequence in the experimental dataset with the lowest Levenshtein distance to all others for AB. For GFP, we set $M = 10$ given that almost all of the sequences in the experimental dataset collected by Sarkisyan et al. (2016) have at most this number of mutations w.r.t. the wild-type sequence. For 5-UTR, we set $M = 50$ as the experimental dataset is collected by randomly mutating a 50-nucleotide-long portion of a wild-type mRNA. For AB, we set $M$ to half of the length of the sequences, i.e. $M = 10$. Second, once we have randomly selected a sequence from the repertoire, if the latter has more than $M$ mutations w.r.t. the base sequence, we randomly select a position to un-mutate, otherwise we randomly select a position to mutate and we replace it with a random amino acid (resp. nucleotide) for protein (resp. RNA) applications. Note that we restrict the set of positions considered for mutations within the set of 50 continuous positions randomly mutated in the original experiment for 5-UTR.

**Algorithm** We provide in Algorithm 1 a thorough description of the MAP-ELITES optimization procedure.

---

**Algorithm 1** MAP-Elites

---

Input: Fitness function $s(\cdot)$, behavior descriptor function $b(\cdot)$, centroids $(b_j)_{1 \leq j \leq J}$, initial population $(x_i^{\text{init}})_{1 \leq i \leq I}$ Population size $B$, number of iterations $N_{\text{iter}}$.

    `# Initialize the cells`
    $c_j \leftarrow \text{null}, j = 1, \cdots, J$
    `# Initialize the population`
    $(x_i)_{1 \leq i \leq I} \leftarrow (x_i^{\text{init}})_{1 \leq i \leq I}$
    **for** $k = 0, \cdots, N_{iter}$ **do**
      **for** $i = 1, \cdots, B$ if $k > 0$ else $I$ **do**
        $\tilde{x}_i \leftarrow \text{random\_mutation}(x_i)$ if $k > 0$ else $x_i$
        `# Find the corresponding cell`
        $j^* \leftarrow \text{argmin}_{j=1,\cdots,J} \|b_j - b(\tilde{x}_i)\|$
        `# Tentatively update the cell`
        **if** $c_{j^*}$ is null or $s(\tilde{x}_i) > s(c_{j^*})$ **then**
          $c_{j^*} \leftarrow \tilde{x}_i$
        **end if**
      **end for**
      `# Sample a new population`
      $(x_i)_{1 \leq i \leq B} \leftarrow \text{sample}((c_j)_{1 \leq j \leq J}, B)$
    **end for**

---

**Hyperparameters.** For all datasets, we run MAP-ELITES for 100,000 iterations with a batch size of $B = 128$. We use repertoires with $J = 2000$ cells, though we observe that the exact value of $J$ has little impact on the performance of the approach, as described in the appendices appendix G. Given that we use more cells than the targeted number of solutions to submit for oracle evaluation, we detail how to downsample from the final repertoire in the next paragraph. We use an ensemble of $L = 18$ scoring functions and use $\beta = 2$ in eq.(1). We use a two-dimensional behavior descriptor (i.e. $d = 2$ in eq.(4)) and use $p = 16,384$ reference sequences that are randomly selected from the offline dataset, though we observe that the exact value of this parameter has a marginal impact on performance, see fig. 7. As initial population population $(x_i^{\text{init}})_{1 \leq i \leq I}$, we use the 128 initial sequences provided to all optimization methods, as detailed in Section **??**. Finally, we use the Centroidal Voronoi Tessellation technique (Vassiliades et al., 2017) to define the behavior descriptor centroids $(b_j)_{1 \leq j \leq J}$.

**Downsampling the** MAP-ELITES **repertoire.** In all experiments, we use a repertoire with 2000 cells even though we have to submit only $N = 128$ sequences for oracle evaluation. To select only 128 sequences at the end of the MAP-ELITES iterations: (1) we create a new MAP-ELITES repertoire with only 128 cells, (2) we use the Centroidal Voronoi Tessellation technique to define 128 behavior descriptor centroids, and (3) we insert the sequences stored in the final MAP-ELITES repertoire into this new repertoire using the same insertion rule already described in algorithm 1. If some of the cells of this new repertoire are empty (which is almost never the case in practice), we randomly select sequences from the final 2000-cell MAP-ELITES repertoire.

## D. Conservative Training of Scoring Functions

In this section, we provide more practical details for training conservative scoring functions using the method developed in Trabucco et al. (2021) and refer to this work for additional information on the topic. The core idea developed in this work is to adjust the training procedure of the scoring functions in order to make them robust to (gradient-based) adversarial attacks. Specifically, Trabucco et al. (2021) combines the conservative term defined in eq.(2) with the $\ell_2$ regression loss as follows:

$$L(\theta) = \mathbb{E}_{(z,y) \sim p_{data}}[\|y - f_\theta(z)\|_2^2] + \alpha(\mathbb{E}_{\mu(x)}[f_\theta] - \mathbb{E}_{(z,y) \sim p_{data}}[f_\theta(z)]), \tag{5}$$

where: (i) $\mu(x)$ is a uniform discrete distribution over adversarial inputs obtained by first sampling points from the offline dataset and running $T = 50$ consecutive gradient ascents steps on $f_\theta$ starting from these initial points with a learning rate

$\eta_{inner}$, and (ii) $\alpha$ is a scalar learned in an online fashion using a Lagrangian formulation with the objective of keeping the overestimation of $f_\theta$ on $\mu(x)$ contained within a hardcoded threshold set to $\tau = 2$. In our experiments, we notice that high values for $\eta_{inner}$ lead to training instabilities as this cause the conservative term defined eq.(2) to diverge, which in turn translates to poor performance in our benchmarks. As a result, we choose to decrease the value of $\eta_{inner}$ reported in Trabucco et al. (2021) and use $\eta_{inner} = 2$ (resp. $\eta_{inner} = 0.01$) for the AB and 5-UTR datasets (resp. GFP dataset) instead of $2\sqrt{KS}$, where $K$ is the number of letters in the alphabet ($K = 20$ for protein problems and $K = 4$ for RNA problems) and $S$ is the sequence length. $\alpha$ is initialized at $0.1$ and is updated with a learning rate of $0.01$ using the Adam optimizer as in Trabucco et al. (2021).

# E. Training Details

**Metrics**  The metrics we report are defined by:

$$
\begin{cases}
\text{Max}\,(\mathbf{x}) & = \max_{i=1,\ldots,N} f(x_i) \\[1em]
\text{Diversity}\,(\mathbf{x}) & = \dfrac{1}{N(N-1)} \displaystyle\sum_{i=1}^{N} \sum_{j=1}^{N} \mathrm{d}(x_i, x_j) \\[1em]
\text{Mean}\,(\mathbf{x}) & = \dfrac{1}{N} \displaystyle\sum_{i=1}^{N} f(x_i) \\[1em]
\text{Novelty}\,(\mathbf{x}) & = \dfrac{1}{N} \displaystyle\sum_{i=1}^{N} \min_{j=1,\ldots,N_{\text{ref}}} \mathrm{d}(x_i, x_j^{\text{ref}}),
\end{cases}
\tag{6}
$$

where $\mathbf{x} = (x_i)_{1 \le i \le N}$ and $\mathrm{d}(x, y)$ is the Levenshtein distance between $x$ and $y$.

**Oracle architecture and training.**  We use the same neural network architecture as the one used in the Design-Bench repository `https://github.com/brandontrabucco/design-bench` (Trabucco et al., 2022) for the 5-UTR dataset. Specifically, our oracle function is composed of an embedding layer followed by a 2-layer residual convolutional neural network, which is itself followed by a self-attention layer and a linear projection. The embedding layer maps all components of the input sequence (which can either be amino-acid letters or nucleobase letters depending on the dataset) to a learned vector of dimension 120, to which we add a sinusoidal positional embedding of the same dimension. Each residual layer is composed of a one-dimensional convolutional layer with kernel size 5 (and parametrized such that the embedding dimension remains unchanged) followed by a layer normalization and a `relu` activation function. The oracle functions are trained using the $\ell_2$ regression loss for 5 epochs with the Adam optimizer, a batch size of 128 sequences, and a learning rate of $10^{-3}$. Compared to Trabucco et al. (2021) and Jain et al. (2022), note that we use the same neural network architecture across datasets for the oracle function as this speeds up inference significantly compared to using Transformer models, which are used in Trabucco et al. (2021) for the GFP dataset, for a small decrease of accuracy. To provide evidence for this, we report the mean squared error, the Spearman's rank correlation coefficient, as well as the Pearson correlation coefficient on the hold-out test sets in table 3. Observe that the Spearman's rank correlation reported in table 3 for the GFP dataset is only 5% smaller than the one reported in Trabucco et al. (2021) with a Transformer architecture. Also note that the Spearman's rank correlation coefficient reported in table 3 is within 1% of the one reported at `https://github.com/brandontrabucco/design-bench/blob/new-api/README.md` for the 5-UTR dataset.

*Table 3.* Accuracy metrics on hold-out test sets for our oracle functions for all datasets.

| DATASET | MSE | SPEARMAN | PEARSON |
|---|---|---|---|
| GFP | 0.221 | 0.793 | 0.881 |
| 5-UTR | 0.168 | 0.887 | 0.912 |
| AB | 0.608 | 0.857 | 0.492 |

**Scoring functions' architecture.**  Following Trabucco et al. (2021) and Jain et al. (2022), our scorers are fully-connected neural networks with 2 hidden layers, 2048 neurons per layer, and with `relu` activations. Just like in Trabucco et al. (2021)

and Jain et al. (2022), the fully-connected neural networks are given as input continuous relaxations of the sequences' one-hot encodings, namely the log of the linear interpolation between a uniform distribution and the actual dirac distribution. Mathematically, the input fed to the neural networks is defined by:

$$\tilde{X} = \log(CX + \frac{1.0 - C}{K}),$$

where $\log$ is applied component-wise, $C$ is a positive constant (which we set to 0.6 as in Trabucco et al. (2021)), $X$ denotes the one-hot encoding vector of a biological sequence, and $K$ is the number of letters in the alphabet ($K = 20$ for protein problems and $K = 4$ for RNA problems). Since the coordinates of $\tilde{X}$ are linearly dependent, a degree of freedom is removed by discarding the last coordinate of $\tilde{X}$ before feeding it to the neural networks.

**Datasets' relabelling.**  For any given dataset, once the oracle function is trained, we relabel the sequences in the dataset with the predictions of the oracle instead of the original experimental measurements. The rationale behind this step is consistency: given that all methods will be evaluated w.r.t. the oracle function and that the latter is not a perfect reflection of the experimental dataset (see table 3), providing experimental measurements in the offline dataset brings additional noise to our benchmarking experiments. In the limit where the experimental dataset includes a non-negligible fraction of all possible sequence options (this is the case for instance for the "TF Bind 8" dataset used in (Trabucco et al., 2022)) and if the oracle function does not fit perfectly the experimental dataset, it might be possible to select sequences that do not perform well w.r.t the oracle function but for which we can attest that they perform well in reality given the experimental dataset. One could argue that in this case we should not even use an oracle function but this would only be possible for truly exhaustive datasets (which are scarce and do not represent the vast majority of biological sequence design problems where the design space is enormous).

**Scoring functions training.**  In accordance with prior work (Trabucco et al., 2021; Brookes et al., 2019; Jain et al., 2022), the offline dataset made available to the optimization methods is only a fraction of the original dataset that does not contain the best performing sequences. For instance, Trabucco et al. (2021) eliminate the top 50% of the sequences ranked by their scores for the 5-UTR dataset and only keep 5000 samples drawn from between the 50th percentile and 60th percentile of proteins in the GFP dataset sorted by score. We believe that restricting the dataset to low-performing samples hampers the interpretability of the optimization results since it is then difficult to attribute poor performance to the optimization method rather than to the poor accuracy of the scoring functions' predictions w.r.t. the oracle specifically for high-performing sequences. However, we also believe that the offline dataset construction should reflect the fact that we are more likely to have found low-performing sequences than high-performing ones during in-vivo experiments. As a compromise, each sample from the full dataset is incorporated into the offline dataset with probability proportional to $\exp^{-f(x)}$, where $f(x)$ is the oracle prediction, with a normalization factor such that the the size of the offline dataset is a third of the full dataset's size. Compared to using a fixed fraction of the dataset, as in Trabucco et al. (2022) for the 5-UTR dataset, this brings the additional advantage that the offline dataset is randomly generated which makes the analysis more robust to the exact definition of the offline dataset as we repeat the optimization experiments a number of times. Scores in the original datasets are normalized between 0 and 1. All scoring functions are trained on the offline dataset using either the $\ell_2$ regression loss or the conservative regression loss eq.(5) for 50 epochs with the Adam optimizer, a batch size of 128, and a learning rate of $3.10^{-4}$.

## F. Hyperparameters for Baseline Optimization Methods

**CbAS.**  We adapt the implementation of CbAS provided by Trabucco et al. (2022) and use the same hyperparameters (Variational Auto Encoder learning rate, batch size, online batch number, epochs) for the GFP and 5-UTR datasets. For the AB dataset, we use a batch size of 32 and 32 online batches during the CbAS iterations due to significant improvements noticed during evaluation. The empiric oracles used to optimize the design consist of an ensemble of conservative scorers whose architecture and training are defined in appendix B. We use the same number of scoring functions for ensembling as MAP-ELITES, see appendix C.

**CMA-ES.**  We use the implementation of CMA-ES provided in the QDAX library (Lim et al., 2022). For all datasets, we set as initial mean the average logits of the starting sequences and we use $10^{-3}$ as initial value for $\sigma$. We follow the procedure developed in Trabucco et al. (2021) for CMA-ES and carry out 100 optimization steps. The scoring function used to optimize the design is the one defined in eq.(1).

**COMS.** We use the same hyperparameters (learning rate, batch size, number of gradient steps) as Trabucco et al. (2021), setting aside the $\eta_{inner}$ parameter defined in appendix D as we have noticed that this parameter has a dramatic impact on the stability of training. That is, if $\eta_{inner}$ is too high, the training of the scoring function becomes unstable and diverges, making the optimization impractical and unreliable. Therefore, we chose the value of $\eta_{inner}$ which we have determined to be the highest value for which the training of the scoring functions remains stable through cross validation. Just like in Trabucco et al. (2021), we also use these values when running gradient ascent on the scoring functions when generating sequences for oracle evaluation. If a significant drop in diversity or in the mean metric is observed, the learning rate is divided by 10.

**GA and GA-HC.** We use the same mutation operators and the same number of scoring functions for ensembling as MAP-ELITES, see appendix C. We run these algorithms for 100,000 iterations with a batch size of 512 sequences. If a drop in diversity is observed, the number of steps is reduced down to 10,000.

**GFLOWNET.** GFlowNet implements a generative network that learns to sample solutions proportionally to the expected reward, see Jain et al. (2022). For all datasets, the networks are trained with a batch size of $64$ and a learning rate of $5.10^{-4}$ for 20 epochs with the Adam optimizer with default values for $(\beta_1, \beta_2)$. The initial value of $\log z$ is set to $50$ since we find that higher values of $\log z$ tend to stabilize training. This parameter is updated with the Adam optimizer with a learning rate of $0.1$ (resp. $5.10^{-3}$) for the GFP dataset (resp. for the AB and 5-UTR datasets). Just like in Jain et al. (2022), we set the $\delta$ parameter to $0.05$, where $\delta$ is the proportion of sequences sampled from the generator to be taken into account in the model loss. The reward used in our model is an exponentiated version of the ensemble loss defined in eq.(1), since theoretical guarantees for this algorithm only stand for positive rewards. Two additional hyperparameters are introduced in the authors' code at `https://github.com/MJ10/BioSeq-GFN-AL`: a sampling temperature and an output multiplicative coefficient. We found that these parameters have a dramatic impact on performance and set the sampling temperature to $0.1$ (resp. $0.5$) for the GFP dataset (resp. AB and 5-UTR datasets) by trial and error. We set the output multiplicative coefficient to $10$ as in the authors' implementation.

**GRAD and GRAD-ENS.** We use the same hyperparameters (learning rate, batch size, number of steps) as for COMS. Just like for COMS, the learning rate is divided by 10 if a significant drop in diversity or in the mean metric is observed. For GRD-ENS, we use the same number of scoring functions as for MAP-ELITES, GA, and GA-HC.

**REINFORCE.** We adapt the implementation of REINFORCE provided by Trabucco et al. (2022), but tweak some of the hyperparameters due to significant improvements noticed during evaluation. Across all datasets, we use a learning rate of $0.001$, batches of size 32 and use 200 REINFORCE iterations. The empiric oracle used to optimize the design is the scoring function defined in eq.(1). We use the same number of scoring functions for ensembling as MAP-ELITES, see appendix C.

## G. Additional Results

An extension version of tables 1, 2 and 4 is provided in table 7 with results for more methods (including ablation studies discussed in appendix G.4).

### G.1. Results for the Antibody Design data

We report in table 4 the results of our algorithm and baselines on the antibody design task.

### G.2. Metrics Evolution and Repertoire Visualisation

In fig. 2, we show evolution of QD metrics during the optimization procedure and on fig. 3 we depict the repertoire fitness at the end of the optimization.

### G.3. Hyperparameters Sensitivity

We conduct an analysis to assess the robustness of optimization methods w.r.t. hyperparameters. All tables and figures are deferred to the Appendix due to space constraints. For MAP-ELITES, we vary the number of cells (resp. the number of reference sequences) from 500 to 5000 (resp. 1024 to 16384) and observe that these parameters have a marginal impact on the metrics (less than 2%). On the contrary, we observe that gradient-based methods are very sensitive to the learning rate and the number of optimization steps. For instance, multiplying the number of steps by a factor 2 for GRAD on the AB

*Table 4.* Mean and standard deviation of metrics from eq.(6) over 5 runs for various methods on the AB dataset.

| METHOD | MAX | MEAN | DIVERSITY | NOVELTY |
|---|---|---|---|---|
| CBAS | $0.55 \pm 0.02$ | $0.34 \pm 0.01$ | $12.9 \pm 0.1$ | $6.38 \pm 0.1$ |
| CMA-ES | $0.53 \pm 0.00$ | $0.43 \pm 0.01$ | $\mathbf{19.0 \pm 0.0}$ | $\mathbf{19.8 \pm 0.1}$ |
| COMS | $\mathbf{0.67 \pm 0.03}$ | $0.52 \pm 0.03$ | $11.3 \pm 0.5$ | $12.0 \pm 0.7$ |
| GA | $0.55 \pm 0.02$ | $0.40 \pm 0.00$ | $13.3 \pm 0.2$ | $6.2 \pm 0.1$ |
| GFLOWNET | $0.41 \pm 0.01$ | $0.28 \pm 0.00$ | $12.6 \pm 0.2$ | $5.7 \pm 0.2$ |
| GRAD | $0.64 \pm 0.02$ | $0.55 \pm 0.02$ | $3.2 \pm 1.2$ | $16.8 \pm 0.3$ |
| REINFORCE | $0.44 \pm 0.03$ | $0.32 \pm 0.02$ | $12.6 \pm 0.8$ | $7.1 \pm 0.6$ |
| **OURS** | $0.66 \pm 0.02$ | $\mathbf{0.56 \pm 0.01}$ | $9.7 \pm 0.4$ | $7.1 \pm 0.3$ |
| OURS-BIO | $0.64 \pm 0.02$ | $0.50 \pm 0.01$ | $12.6 \pm 0.3$ | $8.0 \pm 0.6$ |

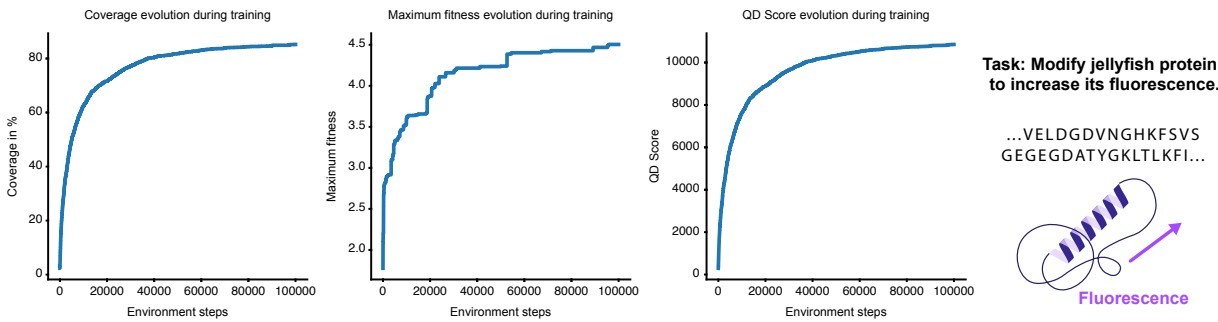

*Figure 2.* Evolution of several QD metrics during optimization: coverage, maximal fitness and QD-score (sum of all the fitnesses in the repertoire) for the GFP redesign task.

dataset results in a significant decrease of the diversity metric (from $3.20 \pm 1.2$ to $0.59 \pm 0.41$). Similarly, multiplying the learning rate by 3 for GRAD on the GFP dataset results in a significant decrease of the mean metric (from $0.75 \pm 0.00$ to $0.67 \pm 0.20$).

We run a series of experiments to identify the sensitivity of the COMS method w.r.t. the learning rate used when running gradient ascent on the scoring functions to generate the pool of sequences for oracle evaluation. Specifically, we run the same experiments described in section 4 on the AB dataset but for four additional values of the learning rate: $l_r/16, l_r/4, l_r *4, l_r *16$, where $l_r$ refers to the value of the learning rate used for COMS in section 4. Results are reported on fig. 4. We observe that the performance of COMS w.r.t. the mean metric is significantly affected by a change in the learning rate (the mean metric drops by at least 20%).

## G.4. Ablation Studies

**Definition of the behavior descriptor.** First, we investigate whether using biologically-grounded behavior descriptors that do not leverage the offline dataset would yield better results than using eq.(4). Specifically, we take the mean polarity, the mean net charge, and the mean hydrophobicity of constituent amino acids as descriptors for the protein datasets (i.e. GFP and AB), resulting in a 3-dimensional behavior descriptor space. For 5-UTR, we use the mean frequencies of three nucleobases within the sequence as 3-dimensional behavior descriptor space (since using all four nucleobases would be redundant). The results of running MAP-ELITES with these alternate behavior descriptors are reported in Tables 1, 2, and 4 where this method is denoted by OURS-BIO. Observe that OURS-BIO tends to achieve higher novelty and diversity metrics but at the price of a significant decrease in the mean metrics for the GFP and AB datasets. More details about this study are provided in the Appendix.

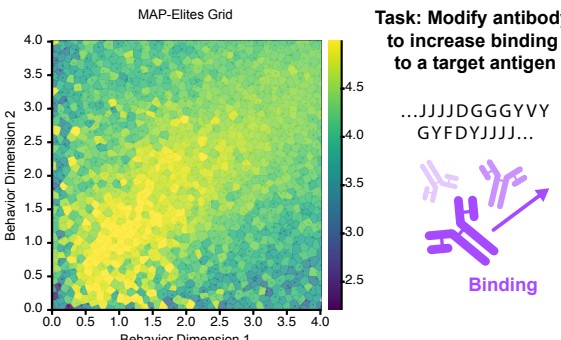

*Figure 3.* MAP-ELITES repertoire at the end of the optimization procedure for the antibody redesign task. The color of each cell indicates the fitness of the best cell's genotype.

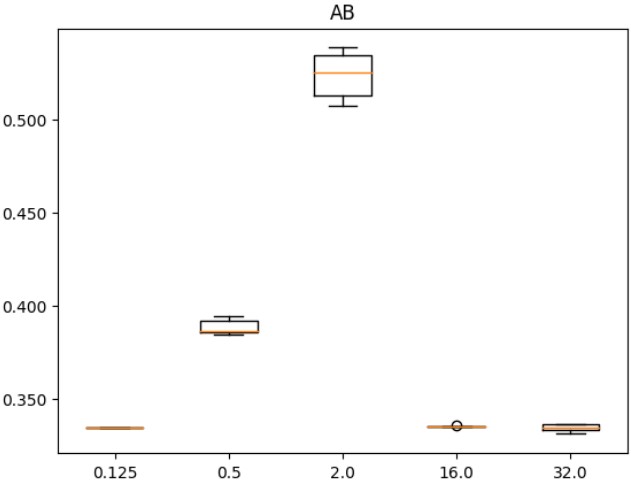

*Figure 4.* Evolution of the mean metric distribution over 8 runs for COMS as a function of the value of the learning rate used to run gradient ascent on the scoring function for the AB dataset (2.0 is the nominal value used in Section 4).

*Table 5.* Mean metrics for our method when fitting scoring functions with the additional loss term eq.(2) versus without.

|         | WITHOUT EQ. (2) | WITH EQ. (2) |
|---------|-----------------|--------------|
| 5-UTR   | $0.69 \pm 0.04$ | $0.69 \pm 0.00$ |
| GFP     | $0.14 \pm 0.06$ | $0.82 \pm 0.01$ |
| AB      | $0.50 \pm 0.00$ | $0.55 \pm 0.01$ |

**Definition of the fitness function.** We now validate the various design choices made in Section 3.2. First, we run an ablation study where we do not include the loss from eq.(2) when fitting scoring functions and observe that this results in a significant decrease in the mean metric on the GFP and AB datasets, see Table 5. Next, we investigate whether using average scores (i.e. setting $\beta = 0$ in eq.(1)) instead of confidence interval lower bounds has any impact on performance and observe that this leads to a non-negligible decrease in the mean metric on the AB dataset. Finally, we vary the number of scoring functions (i.e. $L$ in eq.(1)) from 2 to 18, and observe that ensembling more than 5 functions yields significantly better results on GFP. Full results are in the Appendix.

**Number of scoring functions to ensemble.** In fig. 5, we report the distribution of the mean metric over 8 runs when varying the number of scoring functions used in the definition of the fitness function for MAP-ELITES eq.(1). As can be seen on this figure, this parameter has little impact on the mean performance of our method across datasets as long as it is set to a value larger than 2.

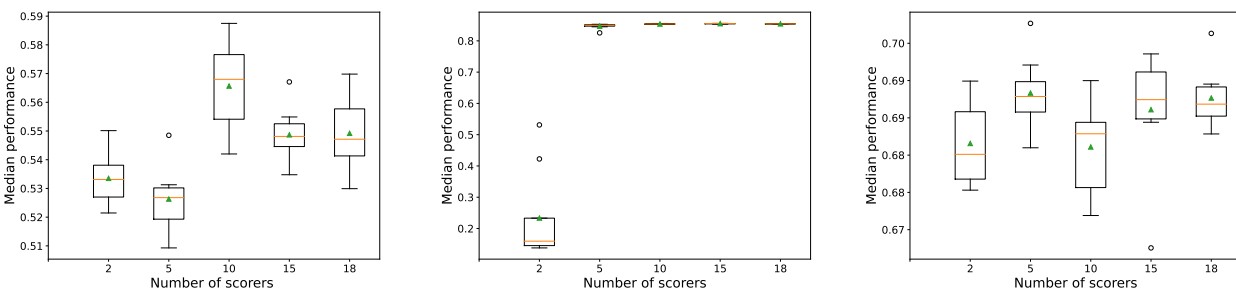

*Figure 5.* Evolution of the mean metric distribution over 8 runs for our method as a function of the number of scoring functions (i.e. $L$ in eq.(1)) on the AB, GFP, and 5-UTR datasets (from left to right).

**Number of cells in the MAP-ELITES repertoire.** In fig. 6, we report the distribution of the mean metric over 8 runs when varying the number of cells (i.e. $J$) in the MAP-ELITES repertoire. As can be seen on this figure, this parameter has little impact on this metric (less than 2%).

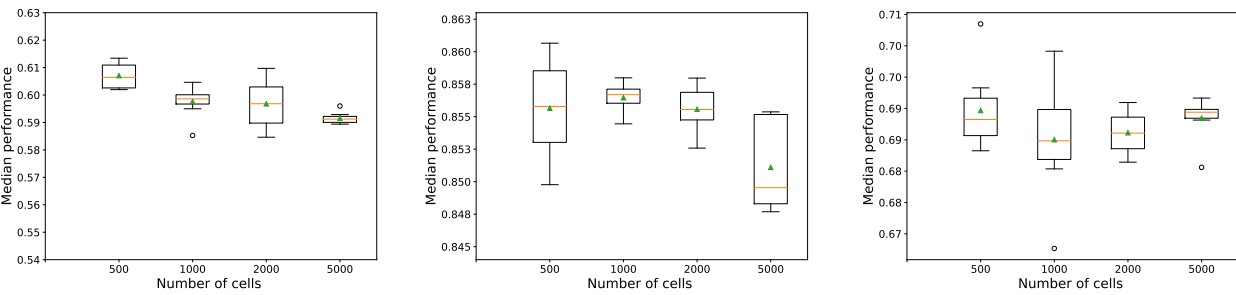

*Figure 6.* Evolution of the mean metric distribution over 8 runs for our method as a function of the number of cells in the MAP-ELITES repertoire on the AB, GFP, and 5-UTR datasets (from left to right).

**Number of reference sequences.** In fig. 7, we report the distribution of the mean metric over 8 runs when varying the number of reference sequences (i.e. $p$ in (3)) used in the definition of the behavior descriptor eq.(4). Again, this parameter has little impact on the performance on the GFP and 5-UTR datasets (less than 1%). The variance of the mean metric distributions reported on fig. 7 for the AB dataset is higher but the medians stay within 3% of the value reported when using 16,340 references.

**Variation of Descriptors.** In addition to random projection and the biologically-grounded behavior descriptors defined in section 5.3 of the main paper, we experimented with a behavior descriptor consisting of a concatenation of the two, and a low-dimensional representation created using an pre-trained Auto Encoder, a method inspired by Cully (2019), which we also call Aurora. The concatenated descriptors yielded interesting results, with most metrics falling between the two MAP-ELITES methods mentioned in sections 5.2 and 5.3, whereas the description given by the Auto Encoder performed significantly worse than either of our methods.

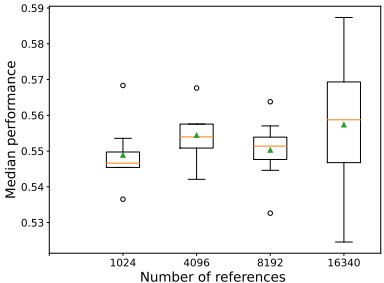 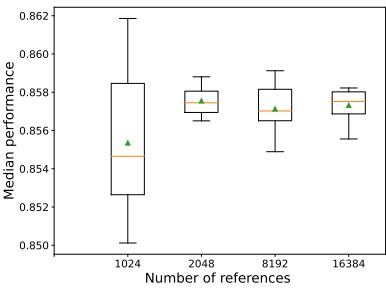 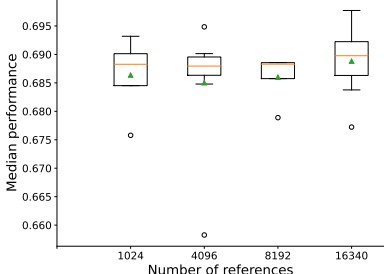

*Figure 7.* Evolution of the mean metric distribution over 8 runs for our method as a function of the number of reference sequences on the AB, GFP, and 5-UTR datasets (from left to right).

*Table 6.* Extension of table 1 with results for more methods. OURS No-COMS refers to our method when fitting scoring functions without the additional loss term defined in eq.(2). OURS $\beta = 0$ refers to our method when setting $\beta = 0$ in eq.(1). Compared to table 1, we report metrics when selecting the entire final MAP-ELITES repertoire, which we denote by the suffix (ALL), as well as metrics when selecting the 128 best-performing (w.r.t. the oracle) sequences from the final MAP-ELITES repertoire, which we denote by the suffix (TOP). Note that this last method could never be implemented in practice since it uses the oracle function and results are provided simply to give more insights.

| DATASET | METHOD | MAX | MEAN | DIVERSITY | NOVELTY |
|---|---|---|---|---|---|
| | CMA-ES | $0.007 \pm 0.083$ | $-0.188 \pm 0.039$ | $232.10 \pm 0.421$ | $199.83 \pm 1.506$ |
| | COMS | $0.855 \pm 0.000$ | $0.751 \pm 0.000$ | $5.859 \pm 0.000$ | $0.075 \pm 0.107$ |
| | GA | $0.858 \pm 0.002$ | $0.800 \pm 0.003$ | $7.899 \pm 0.174$ | $3.742 \pm 0.143$ |
| | GA-HC | $0.857 \pm 0.001$ | $0.795 \pm 0.005$ | $8.820 \pm 0.457$ | $4.745 \pm 0.660$ |
| | GFLOWNET | $0.863 \pm 0.019$ | $0.424 \pm 0.162$ | $86.900 \pm 9.920$ | $110.289 \pm 12.281$ |
| | GRAD | $0.855 \pm 0.000$ | $0.752 \pm 0.002$ | $5.857 \pm 0.005$ | $1.137 \pm 0.714$ |
| | GRD-ENS | $0.855 \pm 0.000$ | $0.751 \pm 0.000$ | $5.859 \pm 0.000$ | $0.075 \pm 0.107$ |
| | OURS (CONCATENATION) | $0.869 \pm 0.062$ | $0.679 \pm 0.106$ | $7.872 \pm 0.651$ | $3.193 \pm 0.391$ |
| GFP | OURS (AURORA) | $0.676 \pm 0.155$ | $0.488 \pm 0.181$ | $9.981 \pm 2.119$ | $9.100 \pm 0.092$ |
| | OURS (ALL) | $0.866 \pm 0.001$ | $0.829 \pm 0.005$ | $7.554 \pm 0.158$ | $3.212 \pm 0.145$ |
| | OURS (N=128) | $0.864 \pm 0.002$ | $0.823 \pm 0.011$ | $8.512 \pm 0.383$ | $4.285 \pm 0.339$ |
| | OURS (TOP) | $0.866 \pm 0.001$ | $0.861 \pm 0.000$ | $8.061 \pm 0.347$ | $4.022 \pm 0.286$ |
| | OURS-BIO (ALL) | $0.869 \pm 0.002$ | $0.402 \pm 0.065$ | $10.503 \pm 0.888$ | $8.151 \pm 0.150$ |
| | OURS-BIO (N=128) | $0.867 \pm 0.002$ | $0.444 \pm 0.081$ | $8.247 \pm 0.888$ | $8.234 \pm 0.186$ |
| | OURS-BIO (TOP) | $0.869 \pm 0.002$ | $0.864 \pm 0.002$ | $8.247 \pm 0.888$ | $8.289 \pm 0.222$ |
| | OURS No-COMS (ALL) | $0.852 \pm 0.012$ | $0.210 \pm 0.064$ | $11.453 \pm 0.362$ | $5.558 \pm 0.306$ |
| | OURS No-COMS (N=128) | $0.712 \pm 0.046$ | $0.177 \pm 0.040$ | $13.299 \pm 0.447$ | $6.367 \pm 0.264$ |
| | OURS No-COMS (TOP) | $0.852 \pm 0.012$ | $0.653 \pm 0.079$ | $23.962 \pm 30.434$ | $4.567 \pm 0.499$ |
| | OURS $\beta = 0$ (ALL) | $0.878 \pm 0.003$ | $0.831 \pm 0.006$ | $10.972 \pm 0.628$ | $7.766 \pm 0.131$ |
| | OURS $\beta = 0$ (N=128) | $0.872 \pm 0.003$ | $0.848 \pm 0.005$ | $9.626 \pm 0.487$ | $8.063 \pm 0.114$ |
| | OURS $\beta = 0$ (TOP) | $0.878 \pm 0.003$ | $0.871 \pm 0.001$ | $8.952 \pm 0.958$ | $8.252 \pm 0.200$ |

*Table 7.* Extension of table 2 with results for more methods. OURS No-COMS refers to our method when fitting scoring functions without the additional loss term defined in eq.(2). OURS $\beta = 0$ refers to our method when setting $\beta = 0$ in eq.(1). Compared to table 2, we report metrics when selecting the entire final MAP-ELITES repertoire, which we denote by the suffix (ALL), as well as metrics when selecting the 128 best-performing (w.r.t. the oracle) sequences from the final MAP-ELITES repertoire, which we denote by the suffix (TOP). Note that this last method could never be implemented in practice since it uses the oracle function and results are provided simply to give more insights.

| DATASET | METHOD | MAX | MEAN | DIVERSITY | NOVELTY |
|---------|--------|-----|------|-----------|---------|
| | CMA-ES | $0.710 \pm 0.000$ | $0.688 \pm 0.000$ | $39.643 \pm 0.047$ | $48.422 \pm 0.035$ |
| | COMS | $0.702 \pm 0.002$ | $0.679 \pm 0.001$ | $36.523 \pm 0.022$ | $1.013 \pm 0.121$ |
| | GA | $0.694 \pm 0.003$ | $0.679 \pm 0.001$ | $25.453 \pm 1.897$ | $11.848 \pm 5.409$ |
| | GA-HC | $0.699 \pm 0.008$ | $0.681 \pm 0.004$ | $4.404 \pm 3.150$ | $12.865 \pm 2.978$ |
| | GFLOWNET | $0.701 \pm 0.053$ | $0.565 \pm 0.065$ | $27.260 \pm 4.215$ | $21.584 \pm 0.998$ |
| | GRAD | $0.696 \pm 0.004$ | $0.648 \pm 0.005$ | $33.941 \pm 0.644$ | $22.474 \pm 0.180$ |
| | GRD-ENS | $0.704 \pm 0.002$ | $0.680 \pm 0.000$ | $36.501 \pm 0.030$ | $1.869 \pm 0.112$ |
| | OURS (CONCATENATION) | $0.670 \pm 0.003$ | $0.667 \pm 0.003$ | $2.800 \pm 0.689$ | $1.400 \pm 0.294$ |
| 5-UTR | OURS (AURORA) | $0.679 \pm 0.003$ | $0.663 \pm 0.003$ | $27.215 \pm (2.760)$ | $14.810 \pm 0.999$ |
| | OURS (ALL) | $0.712 \pm 0.003$ | $0.683 \pm 0.004$ | $16.341 \pm 2.551$ | $22.097 \pm 0.359$ |
| | OURS (N=128) | $0.712 \pm 0.008$ | $0.686 \pm 0.004$ | $14.730 \pm 2.491$ | $22.040 \pm 0.561$ |
| | OURS (TOP) | $0.717 \pm 0.005$ | $0.702 \pm 0.002$ | $14.423 \pm 3.403$ | $22.283 \pm 0.387$ |
| | OURS-BIO (ALL) | $0.710 \pm 0.004$ | $0.665 \pm 0.002$ | $24.728 \pm 1.529$ | $21.931 \pm 0.236$ |
| | OURS-BIO (N=128) | $0.722 \pm 0.000$ | $0.676 \pm 0.002$ | $35.160 \pm 1.607$ | $21.732 \pm 0.258$ |
| | OURS-BIO (TOP) | $0.722 \pm 0.001$ | $0.722 \pm 0.000$ | $50.000 \pm 0.001$ | $21.949 \pm 0.277$ |
| | OURS No-COMS (ALL) | $0.713 \pm 0.007$ | $0.688 \pm 0.002$ | $13.646 \pm 2.445$ | $22.998 \pm 0.351$ |
| | OURS No-COMS (N=128) | $0.708 \pm 0.009$ | $0.688 \pm 0.002$ | $11.056 \pm 2.632$ | $23.041 \pm 0.412$ |
| | OURS No-COMS (TOP) | $0.718 \pm 0.005$ | $0.704 \pm 0.007$ | $11.955 \pm 4.938$ | $23.048 \pm 0.385$ |
| | OURS $\beta = 0$ (ALL) | $0.710 \pm 0.004$ | $0.665 \pm 0.002$ | $24.728 \pm 1.529$ | $19.969 \pm 4.410$ |
| | OURS $\beta = 0$ (N=128) | $0.722 \pm 0.000$ | $0.676 \pm 0.002$ | $35.160 \pm 1.607$ | $19.685 \pm 5.009$ |
| | OURS $\beta = 0$ (TOP) | $0.722 \pm 0.000$ | $0.722 \pm 0.000$ | $50.000 \pm 0.000$ | $20.004 \pm 3.925$ |

*Table 8.* Extension of table 4 with results for more methods. OURS No-COMS refers to our method when fitting scoring functions without the additional loss term defined in eq.(2). OURS $\beta = 0$ refers to our method when setting $\beta = 0$ in eq.(1). Compared to table 4, we report metrics when selecting the entire final MAP-ELITES repertoire, which we denote by the suffix (ALL), as well as metrics when selecting the 128 best-performing (w.r.t. the oracle) sequences from the final MAP-ELITES repertoire, which we denote by the suffix (TOP). Note that this last method could never be implemented in practice since it uses the oracle function and results are provided simply to give more insights.

| DATASET | METHOD | MAX | MEAN | DIVERSITY | NOVELTY |
|---------|--------|-----|------|-----------|---------|
| | CMA-ES | $0.525 \pm 0.002$ | $0.433 \pm 0.005$ | $17.975 \pm 0.047$ | $19.802 \pm 0.109$ |
| | COMS | $0.672 \pm 0.032$ | $0.511 \pm 0.033$ | $10.268 \pm 0.525$ | $11.964 \pm 0.692$ |
| | GA | $0.549 \pm 0.022$ | $0.399 \pm 0.003$ | $13.324 \pm 0.157$ | $6.240 \pm 0.103$ |
| | GA-HC | $0.634 \pm 0.020$ | $0.503 \pm 0.007$ | $7.844 \pm 0.824$ | $7.622 \pm 0.669$ |
| | GFLOWNET | $0.407 \pm 0.010$ | $0.275 \pm 0.002$ | $12.603 \pm 0.212$ | $5.714 \pm 0.185$ |
| | GRAD | $0.639 \pm 0.019$ | $0.558 \pm 0.022$ | $0.589 \pm 0.416$ | $16.830 \pm 0.284$ |
| | GRD-ENS | $0.647 \pm 0.018$ | $0.551 \pm 0.013$ | $8.163 \pm 0.251$ | $13.927 \pm 0.451$ |
| | OURS (CONCATENATION) | $0.643 \pm 0.045$ | $0.531 \pm 0.027$ | $8.715 \pm 0.695$ | $5.089 \pm 0.631$ |
| AB | OURS (AURORA) | $0.516 \pm 0.016$ | $0.468 \pm 0.012$ | $9.391 \pm 0.549$ | $9.353 \pm 0.094$ |
| | OURS (ALL) | $0.697 \pm 0.014$ | $0.546 \pm 0.010$ | $10.201 \pm 0.207$ | $7.635 \pm 0.187$ |
| | OURS (N=128) | $0.662 \pm 0.015$ | $0.555 \pm 0.009$ | $9.655 \pm 0.411$ | $7.086 \pm 0.312$ |
| | OURS (TOP) | $0.697 \pm 0.014$ | $0.635 \pm 0.007$ | $8.788 \pm 0.785$ | $8.219 \pm 0.502$ |
| | OURS-BIO (ALL) | $0.662 \pm 0.008$ | $0.522 \pm 0.007$ | $10.853 \pm 0.400$ | $7.978 \pm 0.628$ |
| | OURS-BIO (N=128) | $0.635 \pm 0.015$ | $0.501 \pm 0.008$ | $12.644 \pm 0.314$ | $8.017 \pm 0.647$ |
| | OURS-BIO (TOP) | $0.662 \pm 0.008$ | $0.605 \pm 0.015$ | $8.202 \pm 1.630$ | $6.735 \pm 1.879$ |
| | OURS NO-COMS (ALL) | $0.689 \pm 0.026$ | $0.502 \pm 0.003$ | $11.001 \pm 0.290$ | $8.564 \pm 0.197$ |
| | OURS NO-COMS (N=128) | $0.659 \pm 0.025$ | $0.504 \pm 0.003$ | $11.264 \pm 0.286$ | $8.614 \pm 0.207$ |
| | OURS NO-COMS (TOP) | $0.689 \pm 0.026$ | $0.605 \pm 0.008$ | $10.888 \pm 0.513$ | $8.768 \pm 0.382$ |
| | OURS $\beta = 0$ (ALL) | $0.688 \pm 0.012$ | $0.520 \pm 0.011$ | $9.586 \pm 0.240$ | $8.287 \pm 0.226$ |
| | OURS $\beta = 0$ (N=128) | $0.657 \pm 0.022$ | $0.526 \pm 0.014$ | $10.041 \pm 0.332$ | $8.323 \pm 0.240$ |
| | OURS $\beta = 0$ (TOP) | $0.688 \pm 0.012$ | $0.623 \pm 0.011$ | $8.874 \pm 1.015$ | $8.352 \pm 0.267$ |