# OpenReview forum: "Quality-Diversity for One-Shot Biological Sequence Design"
_ICML.cc/2024/Workshop/ML4LMS — ML4LMS Poster_

### Official Review · Reviewer_P62q · 2024-06-02
**The authors define a Quality Diversity optimization approach for biological sequence design, apply it to multiple *in silico* optimization tasks, and rigorously compare to relevant benchmarks.**

**Rating:** 8
**Confidence:** 3

**Review:**

The authors define a Quality Diversity optimization approach for biological sequence design, apply it to multiple *in silico* optimization tasks, and rigorously compare to relevant benchmarks. The MAP-Elites algorithm is the basis of this work. Additionally, the authors use an ensemble of oracles to avoid adversarial examples. In particular, they use a high probability lower bound on the ensemble of oracles. They demonstrate their approach on three tasks/datasets:
1. Protein function optimization (GFP)
2. RNA optimization for translation efficiency (5-UTR)
3. Antibody affinity maturation (AB)

They compare their approach to several SOTA baselines from different modeling/optimization approaches and showcase superiority in many cases on the four metrics they evaluate for: max fitness, mean fitness, diversity, and novelty.

My only major suggestion is to add more details regarding the antibody affinity maturation case study. The description of the dataset is alright but some things are not clear. For example, I'm not sure what the message of Figure 3 is. I'm also not sure how in Table 4 the diversity and novelty can reach 19 and higher given that, according to the `Antibody Design (AB) dataset` section the maximum HCDR3 length is 18 residues. Was the length of the HCDR3s being altered beyond the training data? Or perhaps regions outside of the HCDR3 were mutated?

---

### Official Review · Reviewer_w1Cf · 2024-06-11
**Review of Quality-Diversity for One-Shot Biological Sequence Design**

**Rating:** 8
**Confidence:** 3

**Review:**

**Summary** The authors apply the MAP-ELITES algorithm to generate a set of sequences that attempt to maximize both diversity and quality, the ability of the sequence to carryout a certain function. They demonstrate that their approach minorly improves upon relevant baselines, and they include a careful discussion of the design choices of their method.

**Strengths**
* Well-written and compelling.
* Thorough, with relevant baselines included
* While I have seen a number of papers that try to generate a set of diverse but high quality biological sequences, I've not seen this MAP-ELITES algorithm. Therefore, while the authors acknowledge that there are previous works, I believe this is would be novel for many conference participants. Please not that this is not my precise subfield, so I may be wrong in this evaluation.
* I really liked the penalties for OOD data and using the ensemble variance (the high-probability lower bound on the average score)

**Weaknesses**
* As I acknowledged, it may be less novel than I believe as it has many similarities with Trabucco et al (2021).
* The performance is better than other methods in terms of mean and max from the oracle function; however, their method often exhibits worse diversity and novelty. I agree with the authors that high diversity, in of itself, is not good as the most diverse and novel  sequences would just be random garbage. However, in both Tables 1 & 2 there are methods which achieve similar mean/max scores while exhibiting much higher diversity
* There are a handful of small typos

**Opportunities to Improve**
* I would love to see an analysis that focuses just on the N best performing sequences. While the overall diversity is important, the ideal individual sequence would be both high quality and have many mutations from the base sequences.
* The Levenshtein distance should be augmented with BLOSUM substitution scores. Changing a V to a G should not be considered as diverse as an N to a G.
* I think that extensions incorporating protein language model embeddings (ESM-2B) would be exciting as they have been shown to boost performance in a number of protein design tasks. The current network architecture is very lightweight; building your network on top of those would likely result in higher performance and better OOD performance.
* Integration with newer sequence design methods: There are a ton of new sequence design methods, particularly ones leveraging some form of "discrete diffusion" or autoregressive sampling from a PLM. Given the emergence of these methods, the authors should describe how their method could be integrated with these methods.

Overall review: This is thorough and strong paper that absolutely deserves to be included at the workshop. The paper is very well-written and takes an unusual approach to tackle an important and growing problem.

---

### Official Review · Reviewer_xU3b · 2024-06-12
**Review: Quality-Diversity for One-Shot Biological Sequence Design**

**Rating:** 6
**Confidence:** 3

**Review:**

The manuscript addresses the issue of designing and identifying sequences with desired biological activity, which is highly relevant in industry due to the expensive nature of experimental validation. The authors adapt the MAP-ELITES optimization method to generate diverse and high-performing sequences in a single computational step. A particular strength is the comprehensive benchmarking against state-of-the-art techniques, demonstrating narrowly superior performance in terms of mean metrics across two out of the three datasets. I would suggest to the authors to review the antibody task in detail. While CDR H3 can vary almost randomly, depending on the definition used, the first 2 residues may be a lot more conserved, and adapting their algorithm to incorporate this information can help them improve their results in a zero-shot setting.